# Limited N-Glycan Processing Impacts Chaperone Expression Patterns, Cell Growth and Cell Invasiveness in Neuroblastoma

**DOI:** 10.3390/biology12020293

**Published:** 2023-02-11

**Authors:** M. Kristen Hall, Asif Shajahan, Adam P. Burch, Cody J. Hatchett, Parastoo Azadi, Ruth A. Schwalbe

**Affiliations:** 1Department of Biochemistry and Molecular Biology, Brody School of Medicine, East Carolina University Greenville, 600 Moye Boulevard, Greenville, NC 27834, USA; 2Complex Carbohydrate Research Center, University of Georgia, Athens, GA 30602, USA

**Keywords:** neuroblastoma, N-glycans, oligomannose, cell invasiveness, cell growth, N-glycosylation, cell proliferation, N-acetylglucosaminyltransferases, chaperones, oxidative protein folding, cell morphology, cell cycle

## Abstract

**Simple Summary:**

The attachment of sugar residues to proteins is known as glycosylation. Glycosylation is a stepwise process which is significantly altered during cancer and may thus serve as both a therapeutic and/or diagnostic target. However, our understanding of how and which glycan structures contribute to aggressive cancer behaviors is lacking. In this study, we examined how specific N-glycans contribute to aggressive neuroblastoma behavior. Here, we use genetic editing to direct N-glycan expression in neuroblastoma cells, thus enabling us to assign specific N-glycan structures to cellular properties of neuroblastoma. In addition, we also examine how altering the process of N-glycosylation influences the expression of endoplasmic reticulum (ER) folding and transport proteins. It is critical to understand how N-glycosylation processing may influence ER folding and transport protein expression. This study demonstrates that alterations of N-glycosylation influence neuroblastoma progression by diminishing or enhancing cellular properties associated with cancer progression, along with influencing the expression pattern of ER folding and transport proteins.

**Abstract:**

Enhanced N-glycan branching is associated with cancer, but recent investigations supported the involvement of less processed N-glycans. Herein, we investigated how changes in N-glycosylation influence cellular properties in neuroblastoma (NB) using rat N-glycan mutant cell lines, NB_1(-*Mgat1*), NB_1(-*Mgat2*) and NB_1(-*Mgat3*), as well as the parental cell line NB_1. The two earlier mutant cells have compromised N-acetylglucosaminyltransferase-I (GnT-I) and GnT-II activities. Lectin blotting showed that NB_1(-*Mgat3*) cells had decreased activity of GnT-III compared to NB_1. ESI-MS profiles identified N-glycan structures in NB cells, supporting genetic edits. NB_1(-*Mgat1*) had the most oligomannose N-glycans and the greatest cell invasiveness, while NB_1(-*Mgat2*) had the fewest and least cell invasiveness. The proliferation rate of NB_1 was slightly slower than NB_1(-*Mgat3*), but faster than NB_1(-*Mgat1*) and NB_1(-*Mgat2*). Faster proliferation rates were due to the faster progression of those cells through the G1 phase of the cell cycle. Further higher levels of oligomannose with 6–9 Man residues indicated faster proliferating cells. Human NB cells with higher oligomannose N-glycans were more invasive and had slower proliferation rates. Both rat and human NB cells revealed modified levels of ER chaperones. Thus, our results support a role of oligomannose N-glycans in NB progression; furthermore, perturbations in the N-glycosylation pathway can impact chaperone systems.

## 1. Introduction

Neuroblastoma (NB), a pediatric cancer derived from the sympathetic nervous system, accounts for 15% of pediatric cancer-related deaths despite continued improvements in treatment [1]. Non-aggressive/low-risk NB 5-year survival rates exceed 90% and are known to spontaneously regress into benign ganglioneuroma [2]. However, the 5-year survival rate for high-risk NB continues to be under 50%, highlighting the need for improved therapeutic and diagnostic approaches [2,3].

N-glycosylation is a common co- and post-translational modification that is known to be altered in the progression of cancer [3,4,5,6]. The N-glycosylation pathway uses a precursor oligosaccharide that is processed sequentially to produce three general types of N-glycans: oligomannose, hybrid and complex [7], representing the lowest, intermediate and highest amount of glycan processing, respectively. Past studies have supported the role of GnT-V catalyzed β1,6 branched complex glycans in aggressive and metastatic cancers [8,9,10], including NB [3,11]. The expression of hybrid and complex N-glycans, including increases in N-acetyllactosamine (LacNAc) units associated with more branching, support higher amounts of sialyation [12] and fucosylation of N-glycans [13]. In turn, high abundancies of sialylation [14] and fucosylation [15,16] have been linked to metastatic behaviors. However, a recent comprehensive study of breast cancer revealed that higher levels of core fucosylation and lower levels of sialylated N-glycans were associated with more aggressive cancers [17]. Several studies report elevated levels of oligomannose N-glycans in several late-stage/advanced cancer types [17,18,19,20,21,22]. Furthermore, an engineered NB cell line with a virtually sole expression of oligomannose N-glycans had heightened invasive characteristics compared to the parental cell line [23]. 

The chaperone systems and oxidative protein folding in the endoplasmic reticulum (ER) are crucial for the proper folding of secretory soluble and membrane proteins, including glycoproteins [24,25,26,27]. The BiP/Grp94 system consists of chaperones from the heat shock protein (HSP) family [25]. BiP and Grp94 bind and release peptide chains at hydrophobic regions in an ATP-dependent process to guide protein folding [28,29]. The calnexin (CNX)/calreticulin (CRT) cycle is driven by ER lectins, which aid in glycoprotein folding and retention by interacting with their glycans [30,31,32]. The formation/rearrangement of disulfide bonds requires a REDOX reaction that defines the oxidative protein folding process. The protein disulfide isomerase (PDI) family is a group of proteins responsible for breaking and forming disulfide bonds of newly synthesized proteins [27]. Disulfide bond formation/rearrangement conducted by PDIs is aided by Ero1 (ER oxidoreductin) [24,32,33], which regulates the activity of PDI and the ER redox state [33]. Thus, the interplay of ER chaperones, including PDIs (e.g., PDIA1, ERp57) and Ero1 (e.g., Ero1-Lα), is vital for the proper folding of proteins and their exit from the ER [31,33,34]. 

Previously, our lab showed that the editing of *Mgat1* or *Mgat2* in the rat NB_1 (parental) cell line resulted in altered N-glycosylation pathways that produced higher levels of oligomannose and hybrid N-glycans, respectively [23,35,36]. In both NB_1(-*Mgat1*) and NB_1(-*Mgat2*) cell lines, cell proliferation and invasiveness were altered compared to the NB_1 cell line. The loss of *Mgat1* resulted in decreased proliferation, but increased invasiveness, while the loss of *Mgat2* resulted in decreased proliferation and invasiveness. In this study, we engineered a cell line by editing the *Mgat3* gene, called NB_1(*-Mgat3*). Lectin blotting showed that NB_1(-*Mgat3*) expressed lowered bisecting GlcNAc (N-acetylglucosamine) N-glycans relative to NB_1 cells, while cell invasion was similar to NB_1 cells, and cell proliferation was slightly faster. Cell–cell interactions of NB_1(-*Mgat3*) were stronger than NB_1, and furthermore, the cell adhesion properties of NB_1(-*Mgat3*) cells could be rescued. Next, we used electron spray ionization (ESI) mass spectroscopy to characterize the N-glycan structures in the parental NB_1 cells and the mutant cell lines: NB_1(-*Mgat1*), NB_1(-*Mgat2*) and NB_1(-*Mgat3*). The highest level of oligomannose N-glycans was present in NB_1(-*Mgat1*) cells, intermediate in NB_1(-*Mgat3*) and NB_1 cells and lowest in NB_1(-*Mgat2*). Higher levels of oligomannose N-glycans, along with lower levels of fucosylated and sialylated N-glycans, supported a more invasive phenotype, while increased proliferation rates were associated with those cell lines consisting of elevated oligomannose N-glycans with 6–9 Man residues and truncated N-glycans. Western blotting of ER chaperone proteins in the various rat NB cell lines revealed marked differences in protein levels, supporting perturbations in the N-glycosylation pathway impacting the ER chaperone systems, including oxidative protein folding. To expand the roles of N-glycans and chaperones, as well as Ero1 in cell growth and invasiveness, we examined clonal cell lines from human NB (HuNB) SK-N-BE(2): BE(2)-C and BE(2)-M17. Lectin binding studies revealed that higher amounts of oligomannose were associated with the more invasive and slower growing cell line BE(2)-M17. Further levels of chaperone proteins and Ero1-Lα were different between both cell lines. Overall, these results show that oligomannose type N-glycans impact NB progression, and that changes in the N-glycosylation pathway adjust chaperone systems and Ero1-Lα.

## 2. Materials and Methods

### 2.1. Cell Lines and Cell Culture

Human neuroblastoma BE(2)-C, BE(2)C-M17 and Rat B35 neuroblastoma cells were purchased from American Type Culture Collection (Manassas, VA, USA). The rat NB_1 cell line is a clonal cell line derived from the B35 neuroblastoma (NB) cells [23] and was the parental cell line used to engineer NB_1(-*Mgat1*) [23] and NB_1(-*Mgat2*) [35] cell lines. Here, we engineered another rat N-glycosylation mutant cell line, called NB_1(-*Mgat3*). Gene editing of beta-1,4-mannosyl-glycoprotein 4-beta-N-acetylglucosaminyltransferase *(Mgat3)* was accomplished using CRISPR/Cas9 technology. The sgRNA oligonucleotide (ggtggacttcgtgttgccgg) was designed using chop-chop software [37] and the coding sequence included the following nucleotides: 284–304. The accession number of rattus norvegicus *Mgat3* was the following: NM_019239. Recombinant DNA procedures were used to paste the *Mgat3* gRNA into pSpCas9(BB)-2A-Puro vector (Addgene plasmid ID: 48139) [35]. After verifying the correct DNA sequence in the recombinant vector, it was used to silence *Mgat3* in NB_1 cells as previously described [35]. Cell colonies were selected, and fragments of genomic DNA, which encoded the *Mgat3* gRNA sequence, were amplified and cloned into the pCR2.1TOPO vector for DNA sequencing. Sequencing of 9 separate fragments revealed indels in each of them that generated 4 different frameshift mutations: 1 nt insertion, 1 nt deletion, 5 nt deletion and 5 nt indel. Premature stop codons are at nucleotides 420–422 or 470–472 of the coding sequence (CDS) while the next in-frame start sequence occurs at nucleotide 370. This mutated cell colony was amplified and used for experiments. All cells were maintained in DMEM containing 10% FBS, 50 U/mL penicillin and 50 μg/mL streptomycin at 37 °C in a 5% CO_2_ atmosphere. The mouse *Mgat3*-pCDNA3.1-Hygromycin vector kindly provided by Dr. Pamela Stanly, College of Albert Einstein, was used to rescue the NB_1(*-Mgat3*) cell line by employment of the Lipofectamine^®^ 2000 (Thermo Fisher Scientific, Rochester, MA, USA) protocol [36].

### 2.2. Total Membrane Protein Fractions and Whole Cell Lysates

Total membranes of NB cells were isolated via ultracentrifugation as previously described [35]. RIPA buffer (PBS, 1% Triton X-100, 0.5% sodium deoxycholate, 0.1% SDS) with protease inhibitor cocktail set III (EMD Biosciences, San Diego, CA, USA) was used for whole cell lysates [36]. Total membranes and whole cell lysates were mixed in SDS-PAGE sample buffer containing DTT to reduce and denature the samples in preparations for lectin blotting and Coomassie blue staining. Samples were stored at −20 °C until needed.

### 2.3. Western and Lectin Blots 

Proteins of whole-cell lysates were analyzed using lectin blotting, Western blotting and Coomassie-stained gels. Proteins were separated on 10% SDS gels at 20 mA and transferred to PVDF membranes (Millipore, Billercia, MA, USA) at 250 mA for lectin blotting, as previously described [36]. Transferred membranes were incubated with biotin-conjugated E-PHA or GNL lectins (Vector Laboratories, Burlingame, CA, USA). Western blotting was performed using the Bio-Rad ChemiDoc MP Imaging System (Biorad, Hercules, CA, USA) which allowed for the visualization of proteins using fluorescent antibodies. Whole cell lysate samples were electrophoresed on Any kD^TM^ gels (Biorad, Hercules, CA, USA). Separated proteins were transferred to PVDF membranes and were incubated in EveryBlot Blocking Buffer, primary and secondary antibodies, and then imaged using a ChemiDoc MP imaging system. Primary antibodies with rat and human reactivity include: BiP, PDI (PDIA1), calreticulin, Grp94, KIF5B and ERp57 (Cell Signaling Technology, Danvers, MA, USA) and rhodamine conjugated actin (Biorad, Hercules, CA, USA). Primary antibodies with only human reactivity include: Ero1-L-α and calnexin (Cell Signaling Technology, Danvers, MA, USA). The secondary antibody (Biorad, Hercules, CA, USA) used was anti-rabbit IgG starBright Blue 700. Immunobands were quantified using image lab software (Biorad, Hercules, CA, USA) by comparing the ratio of the protein of interest to β-actin, housekeeping protein. In short, multichannel images of Western blots were used to detect protein of interest (starBright Blue 700), along with the housekeeping protein (rhodamine-conjugated β-actin) in each lane. Moreover, multiplexed Western blotting permitted correction for loading and transfer of separated proteins to membranes for each sample since β-actin was measured in the same lane as the protein of interest. The ratio of the immunoband intensities of the protein of interest to β-actin are reported as a mean from at least 3 separate lanes. To verify that proteins of interest and β-actin migrated as reported by the literature which accompanied product, Western blots with protein markers can be viewed in supplementary section (Appendix A). Adjusted band intensity was determined by subtracting local background of immunobands from each immunoband. In all cases, background was kept constant for each Western blot.

### 2.4. Cell Proliferation and Spheroid Invasion

To measure cell proliferation, a colorimetric 5-bromo-2-deoxyuridine (BrdU) assay was used (Millipore, Billerica, MA, USA) following the manufacturer’s protocol and as previously described [36]. Cell proliferation values were determined at an absorbance of 450 nm. Cell spheroids of one to four days old were used for the cell spheroid invasion assay [23]. Cell spheroids were collected and mixed with Matrigel and allowed to invade for 16 h to 20 h. Images were obtained using an Olympus IX73 microscope’s 10× objective to allow for measuring cell spheroid and invasive areas with Image J Software. Cell invasiveness is related to the ratio of invasion area to the sphere area. 

### 2.5. Cell Cycle Assay 

The Premo FUCCI (fluorescence ubiquitination cell cycle indicator) Cell Cycle Sensor Kit (Thermo Fisher Scientific, Rochester, NY, USA) was used to examine the cell cycle of the parental and mutant cell lines. In brief, 5 × 10^4^ cells were plated in 2 mL DMEM media and allowed to incubate for 2 h at 37 °C. After the incubation, the cells were labeled for 16 h in 800 μL labeling volume with Premo CDT1-RFP and Premo geminin-GFP viral particles (80 MOI) as outlined in the accompanying kit protocol. The media were replenished and labeled cells were allowed to incubate for another 8 h before imaging using an Olympus IX-71 microscope (Olympus, Shinjuku City, Tokyo, Japan) with an ORCA R2 deep cooled mono CCD camera. Image J software was used to determine the percentage of cells in S/G2/M phases (geminin-GFP expression), G1 phase (CDT1-RFP expression) and G1-S transition (geminin-GFP and CDT1-RFP expression) per field.

### 2.6. Cell Dissociation Assay 

Cells were grown to confluency on CellBind Culture Dishes (Corning, NY, USA). Cell monolayers were rinsed twice with DMEM and then cells were detached in DMEM using a cell scraper. Detached cells were dissociated by pipetting ten times with a 1 mL pipet tip. Images were obtained of 25–30 fields per dish using a 10× objective from a IX71 Olympus microscope. Areas of cell clusters with more than 10 cells were measured using Image J Software.

### 2.7. Wound Healing Assay 

Cells were plated on 60 mm cell bind dishes (Corning, Corning, NY, USA) and allowed to grow to confluency. Media were removed and wounds were generated using a beveled 200 µL pipette tip (Fisher Scientific, Suwanee, GA, USA). Washing of cellular debris from the plate was carried out twice using 2 mL DMEM. Media were replenished using 3.5 mL of DMEM. Images were obtained at the 0 h and 19 h time points at 4× magnification using an Olympus IX73 Microscope (Olympus, Shinjuku City, Tokyo, Japan). Wound size was measured using Photoshop (AdobePhotoshop, Adobe Inc., San Jose, California, USA) as previously described [35]. Percentage of wound closure = $\frac{\;initial\;wound\;size-final\;wound\;size}{inital}*100$.

### 2.8. N-Glycomics Method 

The total membrane protein fractions extracted from parental and mutant cells were diluted with 100 µL of 50 mM ammonium bicarbonate buffer, followed by the addition of 25 µL of 25 mM Dithiothreitol (DTT) and incubated at 50 °C for 30 min. The samples were subsequently desalted by centrifugal filtration using Amicon centrifuge filters (MilliporeSigma, Cat. No. UFC501096) and then ultrasonicated to dissolve the proteins. The N-glycans were released from the samples by adding 5 µL of PNGaseF (New England Biolabs, Cat. No. P0709L) and incubating at 37 °C for 48 h. The released N-glycans were permethylated by using methyl iodide in the presence of NaOH-DMSO base [38]. The permethylation reaction was quenched by the addition of water and the permethylated N-glycans were extracted with dichloromethane. The dichloromethane layer was washed with water four times, evaporated to dryness by nitrogen gas and the permethylated N-glycans were mixed in 1:1 methanol–water mixture for the ESI-LC-MS/MS analysis. Samples were analyzed on an Orbitrap Fusion Tribrid mass spectrometer equipped with a nanospray ion source and connected to a Dionex binary solvent system (Waltham) as previously described [38]. Prepacked nano-LC columns of 15 cm in length with 75 µm internal diameter filled with 3 µm C18 material (reverse phase) were used for chromatographic separation of the glycans. ESI-LC-MS/MS runs were performed for 72 min gradient using a sodiated buffer system (Buffer A: 1 mM NaOAc in H_2_O, Buffer B: 80% ACN, 0.1% formic acid, 1 mM NaOAc). Precursor ion scan was acquired at 120,000 resolution in the Orbitrap analyzer, and precursors at a time frame of 3 s were selected for subsequent MS/MS fragmentation in the Orbitrap analyzer at 15,000 resolution. MS/MS fragmentation was conducted with fixed CID (Collision Energy 40%) using a data-dependent scan (DDS) program, which performs an MS/MS acquisition for the most abundant ions in the MS^1^ spectrum. Precursors with an unknown charge state, or charge state of + 1 were excluded, and dynamic exclusion was enabled (30 s duration). LC-MS/MS data were analyzed using Thermo Xcalibur 4.2, Thermo FreeStyle 1.7 and GlycoWorkBench software. The area under the curve of different charge states of each N-glycan peak were extracted from the LC-MS chromatogram, added and the relative percentages of individual N-glycans were calculated. The data acquisition and analysis were conducted in duplicate. 

### 2.9. Data Analysis 

The N-glycan structures were assigned with the aid of GlycoworkBench software based on precursor masses (sodium adduct) obtained by ESI-MS/MS and common mammalian biosynthetic pathway [38]. Image J software was used for measurements of neurites, cell colonies, cell spheroids, cell spheroid invasion and cell cycle phases from obtained micrographs. Adobe Photoshop was employed for measurements of cell wound healing assay, and lectin and Western blot pictures. Origin 9.55 was used for graphics and statistics. Data are presented as the mean S.E. where n denotes the number of observations, as indicated. Statistical comparison of two groups was accomplished using unpaired Student’s *t*-test while one-way ANOVA with Bonferroni adjustments was used for more than two groups.

## 3. Results

### 3.1. Characterization of the Newly Engineered NB Cell Lines with an Introduced Mutation in GnT-III

GnT enzymes (*Mgat* genes) are responsible for adding branch points through the addition of GlcNAc residues to the conserved pentasaccharide of N-glycans (Figure 1A). The order for conversion of oligomannose to hybrid and then to complex is initiated by GnTI and followed by GnTII [7]. GnTIII catalyzes the addition of bisecting GlcNAc residues to either hybrid or complex N-glycans to create N-glycans with bisecting GlcNAc residues, terminating the processing of N-glycans [39]. Furthermore, the addition of bisecting GlcNAc residues terminates the processing of N-glycans. To prevent the addition of bisecting GlcNAc residues to N-glycans, the *Mgat3* gene sequence was edited in NB_1, called the parental cell line, by the employment of CRISPR/Cas9 technology. This N-glycosylation mutant cell line is referred to as NB_1(-*Mgat3*). The parental NB_1 cell line is a clonal cell line isolated from rat B35 NB cells [23]. DNA sequencing of nine recombinant vectors which included an amplified genomic DNA fragment of *Mgat3* from a selected cell colony revealed four different indels: a nucleotide insertion and deletion; and two different five nucleotide indels (Figure 1B). In all cases, the indels produced frameshift mutations with premature stop codons. C-Truncated GnT-III proteins would be less than 157 amino acid residues in length, and the sequence following the first 100 residues would be nonsense. The next in-frame start sites could potentially produce N-truncated GnT-III proteins which would contain about 75% of the primary structure of the full-length protein. To verify that GnT-III activity was minimal, glycosylated proteins from whole cell lysates of NB_1(-*Mgat3*) cells, along with those of the parental cell line, were separated on a reducing SDS gel and probed with *Phaseolus vulgaris* Erythoagglutinin (E-PHA) (Figure 1C). This lectin has a high affinity for complex type N-glycans with bisecting N-acetylglucosamine (GlcNAc) relative to other N-glycan structures, and minimal affinity for hybrid type N-glycans with bisecting GlcNAc [40]. The lectin blot showed that E-PHA binding to the isolated glycoproteins of the NB_1(-*Mgat3*) cell line was reduced compared to the parental cell line. The adjacent SDS gel stained with Coomassie blue verifies that the amount of protein loaded per well was similar. Thus, the lectin blot indicated that indels identified by DNA sequences reduced the activity of GnT-III in the NB_1(-*Mgat3*) cell line relative to the parental cell line.

### 3.2. Cell Proliferation Rates and Cell Cycle Phases Were Modified in N-Glycosylation Mutant NB Cell Lines

Previously, it was shown that NB_1 cell lines with introduced indels in *Mgat1* [23] or *Mgat2* [35] genes had much slower growth rates than the NB_1 cell line. Cell proliferation was quantified in sub-confluent cell cultures of NB_1(-*Mgat3*) and NB_1 cell lines by incorporation of BrdU into the DNA during its replication process (Figure 2A). NB_1(-*Mgat3*) cells proliferated slightly faster than NB_1 cells. The NB_1(-*Mgat1*) [23] and NB_1(-*Mgat2*) [35] cell lines had about 50% and 70% reduced proliferation rates relative to NB_1. Taken together, these results demonstrate that more complex-type N-glycans in NB cells promote cell proliferation rates while truncated N-glycans, such as oligomannose and hybrid types, favor slowed rates.

To ascertain whether the different proliferation rates of the N-glycosylation mutant and parental cell lines correlated with changes in cell cycle progression, we performed fluorescent microscopy analysis of these cell lines. Unsynchronized cells were labeled with CDT1-RFP and geminin-GFP, and then micrographs were obtained after 24 h. The percentage of cells in S/G2/M phases (Green), G1 phase (red) and G1-S transition (yellow) of the cell cycle were determined by the expression of geminin-GFP, CDT1-RFP or both fluorescently labeled proteins, respectively, from each micrograph (Figure 2B). The NB_1 and NB_1(-*Mgat3*) cell lines had the largest percentage of cells in the S/G2/M phases and G1-S transition, respectively, while NB_1(-*Mgat1*) and NB_1(-*Mgat2*) had lower levels of cells in these proliferating states (Figure 2C). Moreover, NB_1(-*Mgat1*) and NB_1(-*Mgat2*) had more cells in the nonproliferating phase, the G1 phase. Therefore, a lowered expression of either GnT-I or GnT-II in NB_1 cells reduced the proliferation rate, and the reduction in cell proliferation was due to impeding the progression of cells through the G1 phase of the cell cycle.

### 3.3. Cell–cell Adhesion of NB_1(-Mgat3) Cells Were Rescued by Expression of GnT-III

N-glycan populations of the NB cells influence cell–cell adhesion [23,35,36]. To demonstrate that reducing the activity of GnT-III impacted the ability of NB cells to adhere to each other, a detached cell monolayer was broken apart, and images were obtained of the cell clusters from various NB cell lines (Figure 3A). NB_1(-*Mgat3*) cells formed larger cell clusters than the parental cell line, NB_1. When NB_1(-*Mgat3*) cells were transiently transfected with a mammalian expression vector, containing *Mgat3* cDNA, referred to as NB_1(−/+*Mgat3*) cells, the cell cluster size decreased relative to the NB_1(-*Mgat3*) cell line. Moreover, the cell cluster area of the rescued cell line was 86% of NB_1(-*Mgat3*) and 114% of NB_1 cells (Figure 3B). These results show that enhanced cell–cell adhesion in NB_1 cells with decreased expression of *Mgat3* could be weakened by the ectopic expression of GnT-III.

### 3.4. Reduced Levels of GnT-III in NB Cells Lacked an Effect on Cell Migration and Invasiveness

Past studies showed that reducing the activity of either GnT-I or GnT-II in NB_1 cells had only a slight effect on cell migratory rates of NB cells but a considerable increase and decrease in cell invasiveness, respectively [23,35]. Scratches were introduced into confluent cell monolayers, and micrographs of cell wounds were acquired at 0 h and 19 h. Representative micrographs illustrate the cell wound healing process for NB_1(-*Mgat3*) and NB_1 cell lines (Figure 3C). The quantification of the percentage of cell wound closure from three independent experiments showed that the cell lines had rather similar migratory rates (Figure 3D). Next, the ability of cells to migrate and invade through an extracellular matrix was examined. Spheroids that were 4 day old (Figure 3E,F) were mixed in a Matrigel slurry and plated on cell culture dishes. The spheroids were compact and of similar size (Figure 3G). Micrographs of invading cell spheroids were obtained at 16 h (Figure 3H,I) and 20 h (Figure 3J,K) for NB_1(-*Mgat3*) and NB_1 cell lines. Invading cell spheroids from the NB_1(-*Mgat3*) cell line was much like those of the NB_1 cell line, and in both cases, the invasion was more pronounced when cell spheroids were allowed to invade for a longer period. Since NB cell invasion was dependent on sphere area, cell invasiveness was quantified as the ratio of invasion area to sphere area [23]. The sphere area is enclosed by a white line and the invasion area is within the black line minus the sphere area (Figure 3H). The dependency of the invasion area on the sphere area was supported by Pearson’s correlation values: 0.77, 0.63 for NB_1; and 0.62, 0.72 for NB_1(-*Mgat3*) at 16 h and 20 h invasion periods, respectively. As expected, the cell invasiveness of cell spheroids from NB_1(-*Mgat3*) and NB_1 at the 16 h (Figure 3L) invasion period was less than the cell invasiveness at the 20 h (Figure 3M) period. Hence, compromised activity of GnT-III in NB cells does not appear to alter cell invasiveness, such as reduced activity of GnT-I and GnT-II.

### 3.5. Glycomics Profiling of Parental and N-Glycosylation Mutant NB_1 Cell Lines Were Notably Different

In the current and past studies [23,35,36], types of N-glycans expressed by the parental and N-glycosylation mutant cell lines were based on DNA sequencing, lectin binding studies and glycan attachment of Kv3.1b. To identify specific glycan structures and gain knowledge of how the modifications in the N-glycosylation pathway affect aberrant cellular properties, N-glycans from parental and N-glycosylation mutants were permethylated and isolated, and then analyzed by ESI-MS. The permethylated N-glycans were produced and purified by labeling N-glycosylated proteins from total membranes of various NB cells. The relative abundancies of all glycan structures were determined from the MS^1^ and MS^2^ spectra for each sample performed in duplicate (Appendix A). Averaged and deconvoluted liquid chromatography electrospray ionization mass spectrometry (LC-ESI-MS) spectra of the N-glycans from NB_1 (Figure 4A), NB_1(-*Mgat1*) (Figure 4B), NB_1(-*Mgat2*) (Figure 4C) and NB_1(-*Mgat3*) (Figure 4D) cell lines detected oligomannose, hybrid and complex types of N-glycans in all cell lines; however, the glycan profiles had notable differences. Predominant N-glycan structures for NB_1(-*Mgat1*) and NB_1(-*Mgat2*) were a Man5 oligomannose (structure 4, *m/z* 1580), and a hybrid containing three Man residues (structure 17, *m/z* 2156), respectively. Bisecting GlcNAc N-glycans (structures: 22, 32, 36, 37 39, 40) could be observed in the spectrum for NB_1, while only one of these N-glycans were present in the spectrum of NB_1(-*Mgat3*). Based on the relative abundancies of the N-glycans, both NB_1 and NB_1(-*Mgat3*) cell lines exhibited similar levels (NB_1, 1.2% ± 0.01 and NB_1(-*Mgat3*), 1.3% ± 0.03) of bisecting GlcNAc hybrid (*m/z* 2635, 2809, 2983); however, levels (NB_1, 2.6% ± 0.05; and NB_1(-*Mgat3*), 0.9% ± 0.02) of possible bisecting GlcNAc complex (*m/z* 2081, 2285, 2489, 2646, 2850, 3037, 3211) N-glycans were different. Of note, the bisecting GlcNAc complex N-glycans could not be differentiated from triantennary N-glycans. Thus, these results are consistent with genetic edits and biochemical data according to which GnT-I, -II and -III activities are reduced in the NB_1(-*Mgat1*), NB_1(-*Mgat2*) and NB_1(-*Mgat3*) cell lines, respectively, compared to the parental cell line NB_1.

The quantification of the various N-glycan types showed that the major type of N-glycan was oligomannose but significant differences in the levels of this type of N-glycan were observed (Figure 5A). NB_1(-*Mgat1*) cells expressed the most, NB_1(-*Mgat2*) cells expressed the least and both NB_1 and NB_1(-*Mgat3*) cells had intermediate levels of oligomannose N-glycans. However, NB_1(-*Mgat3*) cells had significantly higher levels than the parental cell line. Hybrid-type N-glycans were also different between the cell lines with the NB_1(-*Mgat2*) cell line expressing higher levels greater than 2-fold. Complex-type N-glycans were more highly observed in the NB_1 and NB_1(-*Mgat3*) cell lines than the other two cell lines with NB_1 cells expressing the highest level. The level of truncated N-glycans was highest in NB_1(-*Mgat1*) cells and lowest in NB_1 cells (Figure 5A, inset). Both NB_1(-*Mgat2*) and NB_1(-*Mgat3*) cells had intermediate levels with the earlier cell line expressing significantly higher levels. The full range of oligomannose N-glycans was observed (*m/z* 1376, 1580, 1784, 1988, 2192, 2396) in the NB cell lines (Figure 5B); and furthermore, all cell lines expressed low levels of paucimannose N-glycans (*m/z* 1346). The overall trend of the various oligomannose structures from Man9 to Man6 displayed a higher expression for NB_1 and NB_1(-*Mgat3*) and a lower expression for the cell lines expressing more truncated N-glycans, NB_1(-*Mgat1*) and NB_1(-*Mgat2*). Moreover, NB_1(-*Mgat1*) had significantly lower levels of Man9, Man8, Man7 and Man6 than NB_1(-*Mgat2*). Although NB_1 had higher levels of Man9 than the NB_1 (-*Mgat3*) cell line, it had significantly lower levels of Man7 and Man6. The levels of Man5 or Man4 structures were similar in cell lines with an active GnT-I: NB_1; NB_1(-*Mgat2*); and NB_1(-*Mgat3*). The summation of percentages of the Man9 to Man6 structures revealed that the expression level was lowest for NB_1(-*Mgat1*) while the NB_1 cell line (-*Mgat2*) was second to lowest (Figure 5C). Both NB_1 and NB_1(-*Mgat3*) cell lines had higher levels with NB_1(-*Mgat3*) expressing the highest level. Differences existed in the levels of fucosylated (Figure 5D) and sialylated (Figure 5E) N-glycans for the various cell lines. The NB_1(-*Mgat1*) cell line produced the least amount of fucosylated and sialylated N-glycans while the NB_1(-*Mgat2*) cell line had the highest levels. Both NB_1 and NB_1(-*Mgat3*) cell lines expressed intermediate levels but NB_1 had significantly more fucosylated and sialylated N-glycans than NB_1(-*Mgat3*). Hence, although all cell lines expressed high levels of oligomannose N-glycans, the N-glycan populations were significantly different. The levels of oligomannose structures detected in the various NB cell lines support that the release of the Man residues in the ER and *cis*-Golgi were slowed in the NB_1 and NB_1(-*Mgat3*) cell lines relative to the other two cell lines. Moreover, the stalling was more apparent in the ER than cis-Golgi for NB_1 than NB_1(-*Mgat3*).

### 3.6. Perturbation in the N-Glycosylation Pathway of NB Cells Modifies Chaperone Protein Levels

To determine whether changes in N-glycan structure expression impacted the ER protein folding process, the abundancies of various chaperones and a protein (KIF5B) assisting in ER exit of folded proteins were evaluated. Whole-cell lysates from confluent cell culture dishes of the various NB cells were employed for Western blotting. In all cases, samples of NB_1, NB_1(-*Mgat1*), NB_1(-*Mgat2*) and NB_1(-*Mgat3*) cells were loaded from left to right. The anti-BiP antibody detected similar levels of BiP in all cell lines (top left panel) while anti-Grp94 (middle left panel), anti-CRT (bottom left panel), anti-ERp57 (top right panel), anti-PDI (middle right panel) and anti-KIF5B (bottom right panel) antibodies revealed differences in their respective immunobands in the various NB cell lines (Figure 6A). Bar graphs below the Western blots show the quantification of each respective protein, as indicated (Figure 6B). BiP levels were not significantly different between the various cell lines. GrP94 levels were nearly 2-fold greater in NB_1 and NB_1(-*Mgat3*) cells than in NB_1(-*Mgat1*) and NB_1(-*Mgat2*) cells. CRT levels were about 1.5-fold higher in NB_1 and NB_1(-*Mgat1*) cells than in NB_1(-*Mgat2*) and NB_1(-*Mgat3*) cells. The NB_1(-*Mgat3*) cell line expressed the highest level of ERp57 followed by NB_1 and then NB_1(-*Mgat1*), while NB_1(-*Mgat2*) expressed the lowest amounts. The expression of PDI was about 1.3-fold higher in NB_1 than NB_1(-*Mgat1*), NB_1(-*Mgat2*) and NB_1(-*Mgat3*) cell lines. KIF5B levels were rather similar in NB_1, NB_1(-*Mgat1*) and NB_1(-*Mgat3*) while those of NB_1(-*Mgat2*) cells were about 2-fold higher. KIF5B expressions of NB_1, NB_1(-*Mgat1*) and NB_1(-*Mgat3*) cell lines were rather similar while the NB_1(-*Mgat2*) cell line expressed about 2-fold higher levels. Thus, these results indicate the levels of proteins in the chaperone systems, including those of the oxidative protein folding process, could be altered due to disruptions in the N-glycosylation pathway.

### 3.7. Human NB Cell Lines with More Oligomannose N-Glycans Have Reduced Cell Proliferation and Enhanced Cell Invasiveness

Previously, our lectin blotting studies showed that glycosylated proteins from the NB_1 cell line bind less Galanthus Nivalis Lectin (GNL) and more L-PHA than the NB_1(-*Mgat1*) cell line, which has an inactive GnT-I [23]. GNL has binding specificity to oligomannose N-glycans [40]. This result was confirmed by the above ESI-MS studies which showed virtually all oligomannose type N-glycans in NB_1(-*Mgat1*). Furthermore, the NB_1(-*Mgat1*) was much more invasive but had slower proliferation rates than NB_1 [23]. Here, the aim was to extend this correlation in two human NB cell lines, (BE(2)-C and BE(2)-M17. GNL interacted in a weaker manner with glycoproteins from BE(2)-C than BE(2)-M17 while the interaction with L-PHA was stronger (Figure 7A). The Coomassie blue-stained gel showed that equal amounts of protein were loaded for the two samples. Cell proliferation was quantified using the incorporation of BrdU into the genomic DNA of sub-confluent BE(2)-C and BE(2)-M17 cell cultures during the replication process (Figure 7B). The proliferation rates were significantly faster for BE(2)-C than BE(2)-M17. Micrographs of the human cell lines at times 0 and 19 h were obtained from cell wound healing assays (Figure 7C). Quantification showed that the cell migratory rates were rather similar for both cell lines (Figure 7D). Micrographs of 1-day old spheroids of BE(2)-C and BE(2)-M17 cells (Figure 7E) were rather uniform but BE(2)-C formed about 1.2-fold larger spheroids than BE(2)-M17 (Figure 7F). Representative images of invading cell spheroids were obtained at 16 h (Figure 7G, upper panels) and 32 h (Figure 7G, lower panels) for BE(2)-C and BE(2)-M17. In both cases, cell invasiveness of 1-day old spheroids of BE(2)-C cells were much reduced when compared to BE(2)-M17 cells after 16 h (Figure 7H, upper panels) and 32 h (Figure 7H, lower panels) of invasion period. As mentioned, NB cell invasion was dependent on sphere area, so cell invasiveness was quantified as the ratio of invasion area to sphere area. The dependency of the invasion area on the sphere are was supported by the Pearson’s correlation values: 0.69 for 16 h invasion; 0.59 for 32 h invasion for BE(2)-C; 0.81 for 16 h invasion; and 0.77 for 32 h invasion for BE(2)-M17. Hence, human NB cells, such as rat NB cells, expressing more oligomannose N-glycans are much more invasive but have a slower proliferation rate.

### 3.8. Distinct N-Glycan Profiles of NB Cells Accompany Changes in the Levels of Chaperone Proteins and Ero1-Lα

The levels of various chaperones were evaluated in human NB cell lines to determine whether differences in N-glycan expression were complemented by changes in the ER chaperone systems and oxidative protein folding. Western blots of whole cell lysates from confluent cell culture dishes of BE(2)-C (lane 1) and BE(2)-M17 (lane 2) NB cell lines (Figure 8A) were performed. Immunoband intensities of BiP, CRT, PDI, ERp57 and Ero1-Lα were higher in BE(2)-C cells than in BE(2)M-17 cells. In contrast, lower and similar immunoband intensities of CNX and Grp94, respectively, were observed for BE(2)-C relative to BE(2)-M17. Quantification of the Western blots of each respective immunoband, as indicated, confirmed these different patterns by their respective protein levels (Figure 8B). PDI levels had the largest difference as BE(2)-C had 3.8-fold more protein than BE(2)-M17. BE(2)-C had about 2-fold higher levels of BiP and CRT than BE(2)-M17. Furthermore, BE(2)-C expressed about 1.5-fold more ERp57 and Ero1-Lα than BE(2)-M17. In contrast, the level of CNX was about half in BE(2)-C compared to BE(2)-M17. Further quantification verified that Grp94 levels were similar in both cell lines. Therefore, these results indicate that the levels of ER chaperone proteins and Ero1-Lα were modified in human NB cell lines with different N-glycan populations.

## 4. Discussion

Our current and past studies [23,35,36] provide a systematic approach to show that disrupting the N-glycosylation pathway affects aberrant cellular properties in NB. Here, we engineered a new NB cell line, NB_1(-*Mgat3*), and characterized it alongside the parental (NB_1) and two previously engineered (NB_1(-*Mgat1*) [23] and NB_1(-*Mgat2*) [35] cell lines. The complement of major N-glycans identified by ESI-MS profiles of all four NB cells were consistent with genetic edits and biochemical data but unexpected high levels of oligomannose N-glycans were detected. Cell proliferation was highest in NB_1(-*Mgat3*) cells, and was followed rather closely by NB_1, while the NB_1(-*Mgat1*) [23] and NB_1(-*Mgat2*) [35] cells were much slower with NB_1(-*Mgat1*) cells proliferating the slowest. Furthermore, the two slowest proliferating cell lines progressed slower through the G1 phase of the cell cycle. Cell invasiveness was highest in NB_1(-*Mgat1*) [23], followed by both NB_1(-*Mgat3*) and NB_1, and then NB_1(-*Mgat2*) [23,35] cell lines. A comparable trend in human BE(2)-C NB cells was observed when silencing *Mgat2,* as the parental cell line was more invasive and proliferated faster [36]. The ranking of the proliferation rates from fast to slow for the parental and N-glycosylation mutant cell lines could be linked to the percentage of oligomannose structure with 6–9 mannose residues from high to low. It was also observed that the faster proliferating cells, NB_1 and NB_1(-*Mgat3*), had less truncated N-glycans than the slower growing cell lines, NB_1(-*Mgat1*) and NB_1(-*Mgat2*), indicating that more processed N-glycans in NB cells had faster cell proliferation rates. Certain N-glycan structures could also be associated with cell invasiveness for the parental and N-glycosylated mutant NB_1 cell lines. Higher oligomannose-type N-glycans, as well as lower fucosylated and sialylated N-glycans, supported the more invasive cell lines, such as NB_1(-*Mgat1*). To expand our findings, we showed that the human BE(2)-C NB cell line which had more complex-type N-glycans was less invasive and had faster proliferation rates than the human BE(2)-M17 NB cell line. Taken together, these results from the parental and mutant NB_1 cell lines, along with those of the two human NB cell lines, contribute significantly to our knowledge of how certain N-glycan structures, such as oligomannose-type N-glycans, promote various phases of NB.

High abundancy of oligomannose N-glycans was identified by ESI-MS spectra in the parental NB cell line. Further GNL interactions of BE(2)-C and BE(2)-M17 cell lines indicated high levels of oligomannose-type N-glycans. These high levels were also suggested in NLF (MYCN-amplified) and SY5Y (MYCN-nonamplified) NB cells with higher levels in NLF cells [11]. It was also shown that higher levels of oligomannose increased the invasiveness of NB cells [23]. Interestingly, the low expression of GnT-V transcripts, which could result in shortened N-glycans, was observed in unfavorable primary NBs [41]. Other cancers have also been shown to be rich in oligomannosidic N-glycans of proteins [17,18,19,20,22]. Specifically, paucimannose N-glycans have been observed as a common component of human cancers [19,22]. Our study showed paucimannose N-glycans present in all four NB cell lines tested, but the NB cell line (NB_1(-*Mgat2*)) with the highest level was the least invasive and proliferated at a relatively low rate [23,35]. Thus, high levels of protein oligomannosylation appear to be a common feature of cancers; however, the components that cause the disruption in N-glycosylation processing is unclear.

Since the processing of N-glycosylated proteins occurs in the secretory pathway, we evaluated ER chaperone proteins in the various NB cell lines. Overall, the N-glycosylation mutant NB_1 cell line had similar or lower levels of the various chaperone proteins to the parental cell line, except NB_1(-*Mgat3*) which had higher levels of ERp57. These results indicate that disruptions in the N-glycosylation pathway were connected to CNX/CRT and GRP78 (BiP)/GRP94 chaperone systems. Further oxidative protein folding was modified in the NB_1 cell line compared to the mutant NB_1 cell line. It should be mentioned that the interaction of ERp57 with CRT or CNX enhances its activity [26], which further suggests the higher requirement for disulfide isomerase activity for proper folding of glycosylated proteins in the parental cell line. Moreover, the increase in isomerase levels could support the higher levels of Man9 and Man8 structures in the NB_1 cell line, as well as the NB_1(-*Mgat3*) cell line. Our examination of human NB cell lines showed that the BE(2)-C cell line which had more and less of the complex and truncated N-glycans (e.g., oligomannose N-glycans), respectively, had modified ER chaperone levels compared to BE(2)-M17. BiP, CRT, PDI, ERp57 and Ero1-Lα levels were raised in BE(2)-C while CNX was lowered. Like NB_1, disulfide isomerase activity appeared to be significantly increased in BE(2)-C as both CRT and ERp57 levels increased relative to BE(2)-M17. Further increased levels of PDI, along with Ero1-Lα, supported higher disulfide isomerase activity as oxidative protein folding involves reoxidation of PDI by Ero1 [27]. Due to changes in the expression of disulfide isomerases and Ero1, and the markedly high levels of oligomannose N-glycans in the parental NB cell lines (NB_1, BE(2)-C and BE(2)-M17), as well as other cancerous cells [17,18,20,21,22], it is arguable that cancer cells have higher levels of protein oligomannosylation due to disulfide bond formation(s) which decrease the rate of mannose removal from the oligomannose N-glycans and consequently their conversion to higher-order N-glycans.

Our current study showed varied expressions of ER chaperones in rat NB cell lines with disruptions in the N-glycosylation pathway due to genetic editing, and the human NB cell lines with different N-glycosylation populations acquired during tumor growth. Calreticulin was more abundant in BE(2)-C, while calnexin was more abundant in BE(2)-M17. Moreover, the enrichment of calreticulin in BE(2)-C was accompanied with more BiP and ERp57. This trend was like a previous study that reported cells depleted of calnexin was associated with increased levels of calreticulin, ERp57 and BiP [42]. Calreticulin levels were increased in NB_1 and NB_1 (-*Mgat1*) cells relative to NB_1 (-*Mgat2*) and NB_1 (-*Mgat3*) cells, and in the case of NB_1 the higher calreticulin levels were associated with increased ERp57. Since the ER chaperone network utilizes a compensatory relationship between the multiple families of chaperones to ensure efficient and accurate protein folding [28,43], it may be that directly altering GnT-I activity results in a different type of compensation. ESI-MS spectra revealed that the editing of *Mgat1*, *Mgat2* and *Mgat3* caused a reduction in the expression of GnT-I, GnT-II and GnT-III in the NB_1(-*Mgat1*), NB_1(-*Mgat2*) and NB_1(-*Mgat3*) cell lines, respectively. NB_1(-*Mgat1*) cells expressed at least 98% of oligomannose N-glycans and the most prevalent oligomannose was the Man5 structure. Furthermore, the level of the Man4 structures was quite like the Man8 and Man7 structures, indicating that the glycoproteins with Man5 structures are moving through the secretory pathway and not retained. The low levels of hybrid and complex N-glycans are supported by previous work which showed that the deletion of 120 amino acids from the N-terminus of GnT-I causes a complete loss of activity [44]. As mentioned, the mutated Mgat1 gene in the NB_1(-*Mgat1*) cell line does not have an in-frame start codon until amino acid 251. NB_1(-*Mgat2*) cells expressed the highest levels of hybrid N-glycans and the most abundant hybrid N-glycan structure had less than four Man residues; however, close to 10% of the N-glycans were of complex type. Since GnT-II uses hybrid N-glycans with the α1–3 mannose removed [7], the glycan profile of NB_1(-*Mgat2*) supports that the indel introduced in the *Mgat2* gene reduces GnT-II activity. The N-glycan profiles of NB_1(-*Mgat3*) cells supported changes in the N-glycosylation pathway as the bisecting GlcNAc complex N-glycans were reduced in NB_1(-*Mgat3*) compared to NB_1. As such, it appears the indel in *Mgat1* almost completely silenced the gene, while indels in *Mgat2* and *Mgat3* caused some reduction in the activity of GnT-II and GnT-III. Alternative start codons are the likely reason for the remaining activities of GnT-II and GnT-III. In both the *Mgat2* and *3* genes, there are start codons downstream of the sequenced indels that could serve as alternative start sites at nucleotide positions 292 and 370 of their coding sequences, respectively. N-Truncated proteins of 345 and 415 amino acid residues could be produced for GnT-II and GnT-III, instead of their full-length forms, 442 and 538 amino acid residues, respectively. However, these alternative start sites would be less favored for protein translation [45]. In both cases, the N-truncated GnT-II and GnT-III proteins could retain primary sequences of their C-terminal catalytic domains [46,47]. Taken together, the N-glycan populations of the various mutant cell lines support genetic edits that cause at least a partial reduction in the activity of their respective GnT.

The fucosylation and sialyation of N-glycosylated proteins are reported to be increased in multiple cancers and are thought to contribute to cancer aggressiveness [12,13,15,16]. Our study in NB was in discrepancy with those reports as NB_1 (-*Mgat1*) cells were the most invasive, and they had minimal fucosylation and sialyation while the least invasive cell line, NB_1 (-*Mgat2*), had the greatest levels of fucosylation and sialyation. NB_1 and NB_1 (-*Mgat3*) cells represented intermediate cell invasiveness and levels of fucosylation and sialyation. Since we suspected that the populations of complex glycans were underrepresented, fucosylation and sialyation were also accounted for on truncated N-glycans which gave a similar pattern. Overall, this highlights the complexities of N-glycans in cancer progression and indicates that high fucosylation and sialyation levels may suppress NB cell invasiveness.

Rapid proliferation is a common feature of cancerous cells which makes them prone to ER stress [48]. Faster proliferating cells require increased protein synthesis and enhancement of the N-glycosylation pathway. Elevated levels of both BiP [49,50] and PDI [51] have been associated with cancer progression. Our results are in agreement with those prior studies as increased cell proliferation rates in conjunction with more abundant BiP and PDI were noted in BE(2)-C relative to BE(2)-M17. However, results from the parental and N-glycosylation mutant NB cell lines disputed those claims since similar levels of BiP were detected among the cell lines even though variability was noted in proliferation rates. Furthermore, PDI was more abundant in NB_1 even though NB_1(-*Mgat3*) proliferated faster. As such, when evaluating ER stress via chaperone systems, the significance of disruptions in the N-glycosylation pathway must not be neglected.

## 5. Conclusions

In summary, we present a simplified approach correlating cell invasiveness and proliferation, along with the expression of ER chaperone proteins and Ero1, with changes in the N-glycosylation pathway by editing *Mgat* genes in a rat NB_1 cell line. Furthermore, two clonal cell lines of the SK-N-BE(2) human cell line (BE(2)-C and BE(2)-M17) were implemented to parallel the rat NB cell lines. Specifically, this study revealed a functional role of oligomannose glycans in cell invasiveness and proliferation, thus impacting the progression of NB. Additionally, our studies supported that lowered levels of fucosylated and sialylated N-glycans increased cell invasiveness. Moreover, a correlation between the N-glycosylation pathway and the ER protein folding process via altered ER chaperone expression, along with Ero1, was noted. Our systematic approach clearly showed that differences in the N-glycosylation pathway resulted in modified cell proliferation and invasion. We propose that increased levels of various oligomannose-type N-glycans promote NB aggression.

## Figures and Tables

**Figure 1 biology-12-00293-f001:**
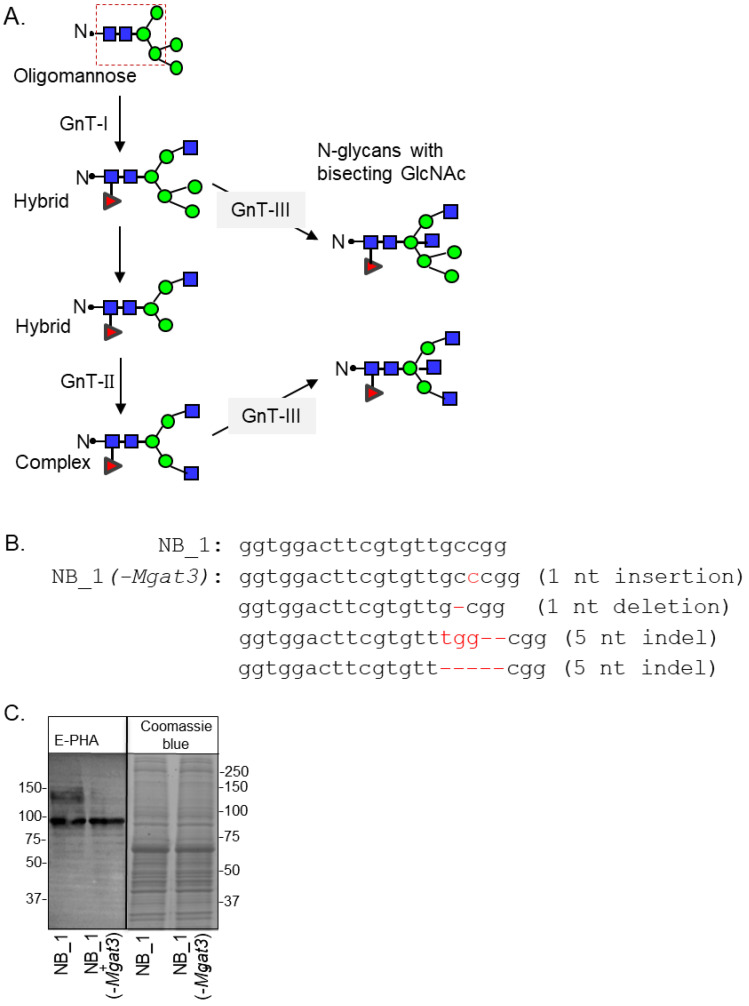
Engineering the knockout of the *Mgat3* gene in the NB_1 cell line. Depiction of the N-glycan processing pathway, in which simple, oligomannose, N-glycans are processed into mature hybrid and complex N-glycans. Dashed box denotes the common core sequence of N-glycans. Oligomannose N-glycans are converted into hybrid via GnT-I, and hybrid N-glycans are converted into complex type through GnT-II activity. GnT-III inserts a bisecting GlcNAc residue on either hybrid or complex, halting further branching of the N-glycan structure (**A**). The coding sequence of rat *Mgat3* from 284 to 304 was compared to that of a newly created NB_1(-*Mgat3*) cell line (**B**). Red hyphens indicate deleted nucleotides. E-PHA lectin blot and Coomassie blue-stained gel of whole cell lysates from NB_1 and NB_1(-*Mgat3*) (**C**). Number adjacent to the blot and gel denote protein markers (in KDa). Full lectin blot and Coomassie blue-stained gel (Appendix A).

**Figure 2 biology-12-00293-f002:**
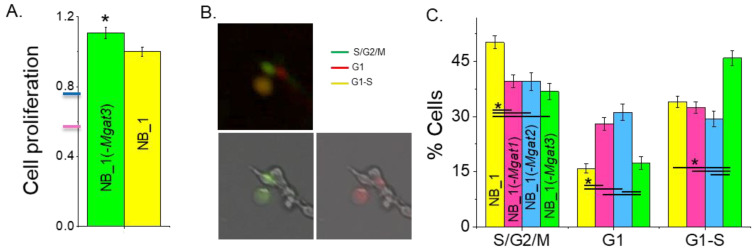
N-glycan diversity impacts cell cycle progression, thus altering NB cell proliferation. Cell proliferation of NB_1 and NB_1(-*Mgat3*) cell lines (**A**). Data are shown as the mean ± SEM (*n* = 7) and were compared by Student’s *t*-test (* *p* < 0.03). Pink and blue tick marks indicate proliferation rates of NB_1(-*Mgat1*) or NB_1(-*Mgat2*), respectively. Representative micrographs of FUCCI stained cells (**B**). Green fluorescing cell (lower left panel), red fluorescing cell (lower right panel) and those images overlaid (upper panel) indicate cell populations in various phases of the cell cycle. Average percent of cells in S/G2/M, G1, or G1-S ± SEM for NB_1 (*n* = 170), NB_1(-*Mgat1*) (*n* = 142), NB_1(-*Mgat2*) (*n =* 110) and NB_1(-*Mgat3*) (*n* = 120) (**C**). One-way ANOVA with Holm–Bonferroni mean comparison (* *p* < 0.05).

**Figure 3 biology-12-00293-f003:**
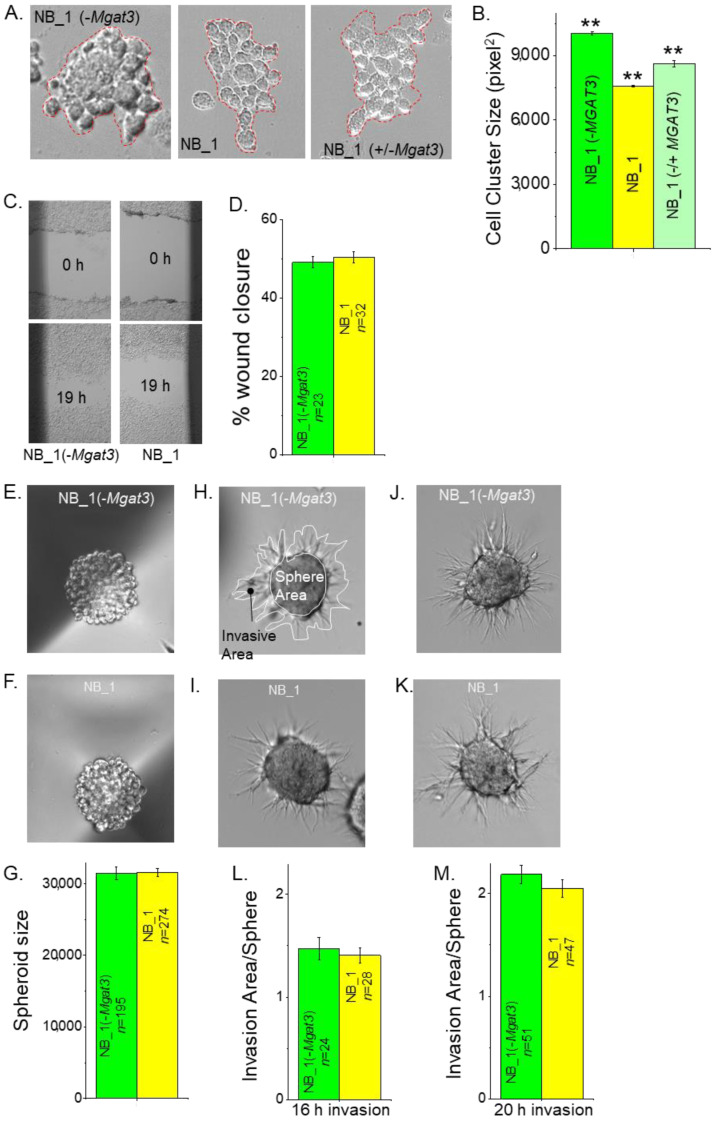
Influence of N-glycan populations on cell–cell adhesion, cell migration and cell invasion. Representative micrographs of resulting cell clusters from dissociated NB_1(-*Mgat3*) (left panel), NB_1 (middle panel) and NB_1(−/+*Mgat3*) (right panel) cell monolayers (**A**). Cell aggregates of more than 10 cells, as encircled in red, were analyzed. The average area of cell aggregates from NB_1(-*Mgat3*) (*n* = 775), NB_1 (*n* = 907) and NB_1(−/+*MGAT3*) (*n* = 775) (**B**). n represents a cell cluster. One-way ANOVA followed by Holmes–Bonferroni’s test was used to compare differences in mean values. A value of *p* < 0.01 was considered significant (**). Images of scratches in cell monolayers of NB_1 and NB_1 (-*Mgat3*) at 0 h and 19 h migration (**C**). The bar graph shows the mean percentage of wound closure at 19 h (**D**). Characteristic images of spheroids formed in culture at 4 days by NB_1(-*Mgat3*) (**E**) and NB_1 (**F**), and their mean spheroid sizes (**G**). Matrigel invasion was captured at 16 h (**H**,**I**) or 20 h (**J**,**K**). Encircled areas in white invasion area or sphere area, respectively (**G**). Bar graphs indicate spheroid invasiveness at 16 h (**L**) or 20 h (**M**). Data are shown as the mean ± SEM and were compared by Student’s *t*-test. n denotes number wounds, spheroids and invading spheroids.

**Figure 4 biology-12-00293-f004:**
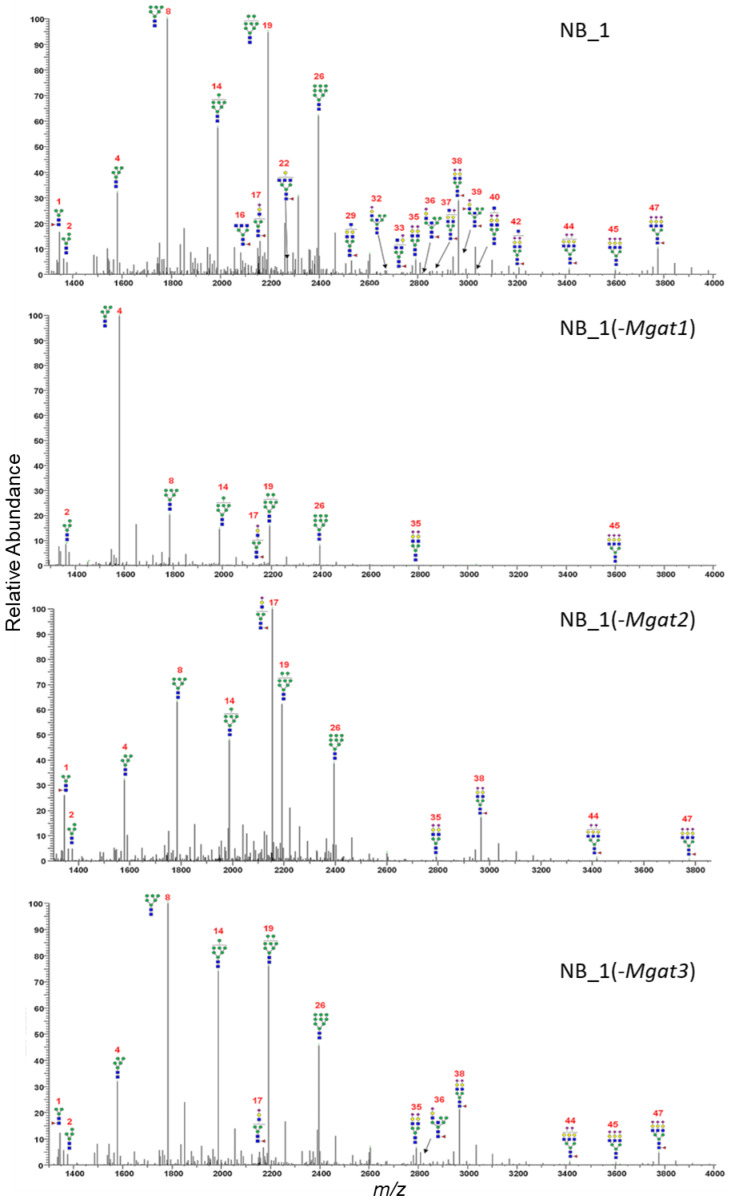
Averaged and deconvoluted LC-ESI-MS spectra of N-glycans released from NB_1, NB_1(-*Mgat1*), NB_1(-*Mgat2*) and NB_1(-*Mgat3*), cell lines. Data were acquired from dichloromethane extracted permethylated N-glycans and all molecular ions are present in sodiated form ([M + Na]+). N-glycans are labelled with numbers, corresponding to Appendix A.

**Figure 5 biology-12-00293-f005:**
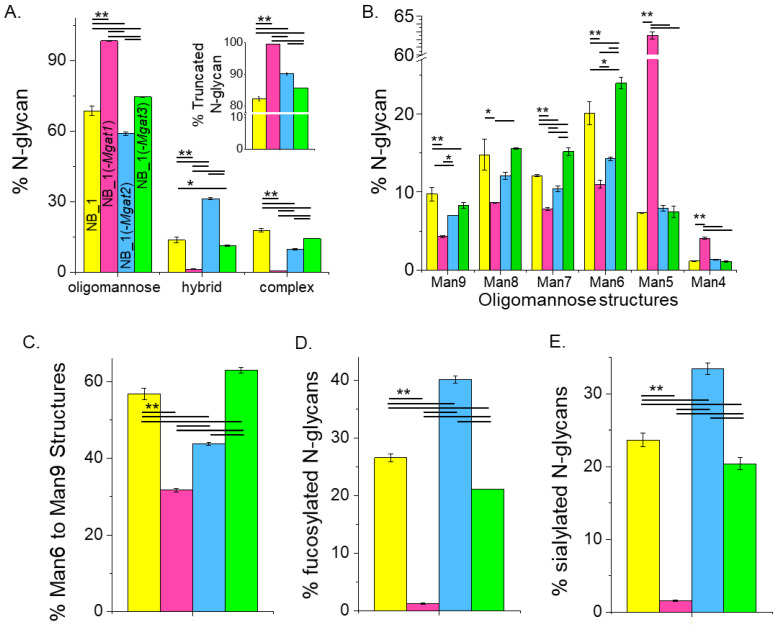
Relating oligomannose, fucosylated and sialylated N-glycans to changes in the N-glycosylation pathway of the various NB_1 cell lines. Summarization of the percent of various N-glycan types per cell line (**A**), large to small oligomannose N-glycans (**B**), 6–9 mannose residues (**C**), fucosylated N-glycans (**D**) and sialylated N-glycans (**E**) as identified by the ESI-MS spectra. The inset (**A**) shows the percentage of truncated N-glycans. Data are shown as the mean ± SEM, *n =* 2, and were compared by one-way ANOVA using Holm–Bonferroni post hoc test (* *p* < 0.1), (** *p* < 0.05).

**Figure 6 biology-12-00293-f006:**
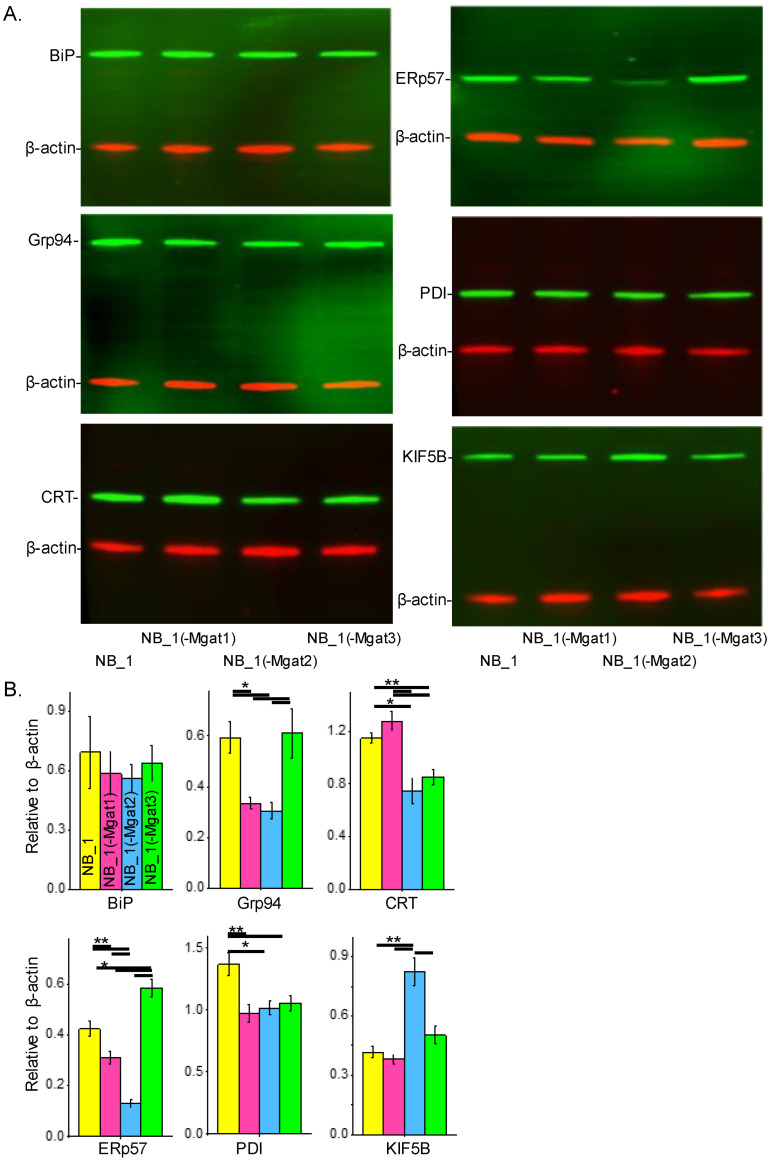
N-glycans and expression of ER chaperones in NB cells. Western blots of BiP (~78 KDa), Grp94 (~94 KDa), CRT (~55 KDa), PDI (~57 KDa), ERp57 (~57 KDa) and KiF5B (110 KDa) multiplexed with β-actin:rhodamine (**A**). In all cases, immunobands of interest were normalized to β-actin for each sample load, and then means were calculated. Bar graphs reveal the level of BiP (*n =* 5), Grp94 (*n =* 4), CRT (*n =* 4), ERp57 (*n =* 4), PDI (*n* = 5) and KIF5B (*n* = 4) expression in NB_1, NB_1(-*Mgat1*), NB_1(-*Mgat2*) and NB_1(-*Mgat3*) cells (**B**). Data are presented as mean ± SEM and were all compared by one-way ANOVA with Holm–Bonferroni mean comparison (** *p* < 0.01, * *p* < 0.05). Full Western blots (Appendix A).

**Figure 7 biology-12-00293-f007:**
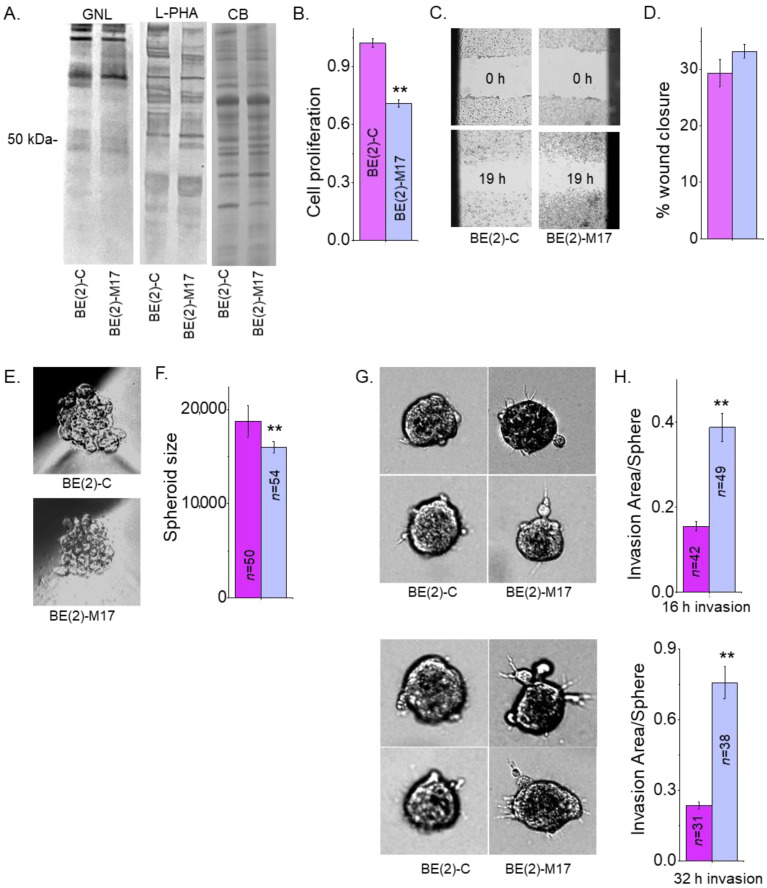
Less processed N-glycans influences human neuronal cell migration and invasion. GNL and L-PHA lectin binding of whole cell lysates from BE(2)-C and BE(2)-M17, and associated Coomassie blue-stained gel of protein loads (**A**). Bar graphs indicate cell proliferation rates of BE(2)-C (*n* = 13) and BE(2)-M17 (*n* = 13) (**B**). Images (**C**) and bar graphs (**D**) depict percentages of wound closure at 16 h in BE(2)-C (*n* = 10) and BE(2)-M17 (*n* = 30) cells. Micrographs of 1-day old spheroids (**E**) and their quantification (**F**). Representative images of spheroid invasion at 16 h ((**G**), upper panels) and 32 h ((**G**), lower panels). Bar graphs illustrating cell invasiveness at 16 h ((**H**), upper graph) and 32 h ((**H**), lower graph) incubation periods for BE(2)-C and BE(2)-M17 cells. Data are presented as mean ± SEM and were all compared by Student’s *t*-test (** *p* < 0.01). Full lectin blots (Appendix A).

**Figure 8 biology-12-00293-f008:**
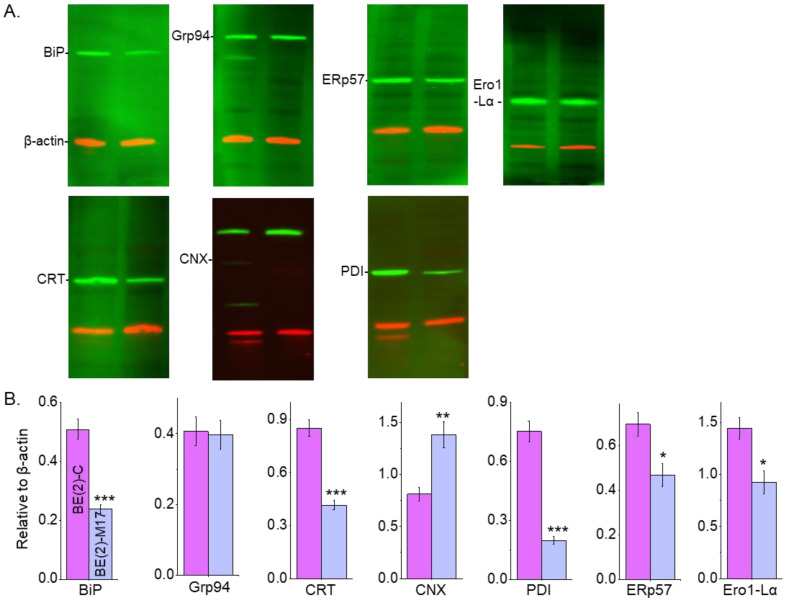
ER chaperones and N-glycan populations of human neuronal cells. Western blots of BiP (~78 KDa), Grp94 (~94 KDa), CRT (~55 KDa), CNX (~90 KDa), PDI (~57 KDa), ERp57 (~57 KDa) and Ero1-L alpha (~60 KDa) multiplexed with β-actin:rhodamine (**A**) and quantification of BiP (*n =* 5), Grp94 (*n =* 6), CRT (*n =* 11), CNX (*n* = 7), PDI (*n* = 6), ERp57 (*n =* 6) and Ero1-L alpha (*n* = 4) in BE(2)-C and BE(2)-M17 cells (**B**). In all cases, immunobands of interest were normalized to β-actin for each sample load, and then means were calculated. Data are presented as mean ± SEM and were all compared by Student’s *t*-test (*** *p* < 0.001, ** *p* < 0.05, * *p* < 0.1). Full Western blots (Appendix A).

## Data Availability

Data is available upon request.

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
