# Peer review of "Limited N-Glycan Processing Impacts Chaperone Expression Patterns, Cell Growth and Cell Invasiveness in Neuroblastoma"

_biology, 2023, doi:10.3390/biology12020293_

Round 1

Reviewer 1 Report

Schwalbe and coworkers presented a manuscript entitled: “Limited N-Glycan processing impacts chaperone expression patterns, cell growth and cell invasiveness in neuroblastoma”. The topic is interesting, experimental work well conducted and presented and conclusions as well. The manuscript can then be published after the authors have responded to the following remarks.

Figure 4 does not account for the differences of the four profiles. Changes are much more evident in the supplementary figures. I would propose to replace figure 4 by adapting supplementary figures S1-S4 and drawing the structures of the more intense peaks. All structures can be included in one or more of the supplementary tables

Figure S8 caption is wrong!

What logic was used in choosing the MS/MS spectra of the Supplementary Figures? Being a work in which NB_1(-Mgat3) cells are used, I would have expected MS/MS of bisected N-glycans.

As regards the bisected structures, they are certainly present at m/z 2635.3, 2809.4 (hybrid structures), while the peaks at m/z 2081.1, 2285.2, 2485.3, 2635.3, 2850.4, 3037.5 and 3211.6 may correspond to triantennary or bisected N-glycans or both.

Are the authors able, through chromatography, to distinguish and give a certain assignment?

These latter peaks are of lower intensity in the spectra in Figure 4d and in Figure S4 (NB_1(-MGat3)cells) as one would expect for bisected N-glycans while the bisected hybrid structures (which are formed in the CIS-Medial Golgi, prior to action of MGat2) have the same intensity as NB_1.

Did the authors notice this strange pattern? Do they have any explanations?

Author Response

Reviewer 1

Authors: We greatly appreciate your comments and suggestions as the manuscript has been strengthened.

Comments and Suggestions for Authors

Schwalbe and coworkers presented a manuscript entitled: “Limited N-Glycan processing impacts chaperone expression patterns, cell growth and cell invasiveness in neuroblastoma”. The topic is interesting, experimental work well conducted and presented and conclusions as well. The manuscript can then be published after the authors have responded to the following remarks.

Figure 4 does not account for the differences of the four profiles. Changes are much more evident in the supplementary figures. I would propose to replace figure 4 by adapting supplementary figures S1-S4 and drawing the structures of the more intense peaks. All structures can be included in one or more of the supplementary tables

Authors: As suggested, figure 4 spectra were substituted with edited spectra from the supplementary section which more clearly show how the bisecting N-glycans are different between the NB_1 and NB_1(_Mgat3) cell lines, and furthermore more clearly show knock-down of Mgat1 and Mgat2 in their respective cell lines. The figure legend has been updated. A description of revised figure 4 has been edited in the text of the results and discussion to support changes in bisecting GlcNAc N-glycans in the mutant cell line relative to the parental cell line.

Figure S8 caption is wrong!

Authors: This has been replaced to the following: Representative ESI-MS2 spectrum of N-glycans released from NB_1(-Mgat3) with mass m/z 2966.5 showing the glycan fragments which confirms the structure.

 What logic was used in choosing the MS/MS spectra of the Supplementary Figures? Being a work in which NB_1(-Mgat3) cells are used, I would have expected MS/MS of bisected N-glycans.

Authors: Those structures that were detected in higher amounts for oligomannose, hybrid and complex type N-glycans were selected for MS/MS spectra.

As regards the bisected structures, they are certainly present at m/z 2635.3, 2809.4 (hybrid structures), while the peaks at m/z 2081.1, 2285.2, 2485.3, 2635.3, 2850.4, 3037.5 and 3211.6 may correspond to triantennary or bisected N-glycans or both.

Are the authors able, through chromatography, to distinguish and give a certain assignment?

These latter peaks are of lower intensity in the spectra in Figure 4d and in Figure S4 (NB_1(-MGat3)cells) as one would expect for bisected N-glycans while the bisected hybrid structures (which are formed in the CIS-Medial Golgi, prior to action of MGat2) have the same intensity as NB_1.

Did the authors notice this strange pattern? Do they have any explanations?

Authors: As mentioned above, the figure 4 was substituted as requested, see above. Of note, we have mentioned these structures and commented about the pattern in the text of the results, and discussion.

We were not able to conduct this study.

In short, the patterns suggest that the NB_1(- Mgat3) cell line has lowered GnT-III activity. Due to the potential of greater diversity of complex than hybrid types of N-glycans in NB_1(- Mgat3) cells, the reduced activity of GnT-III would likely have a greater effect on the complex N-glycans. 

Reviewer 2 Report

Hall and coworkers studies how changes in N-glycosylation influence cellular properties in NB using rat N-glycan mutant cell lines, NB_1(-Mgat1), NB_1(-Mgat2), and NB_1(-Mgat3), and the parental cell line, NB_1 in an efficient manner. They also studied the correlation between invasion property and type of N-glycan present in it and finally looked for change in ER chaperones expressions through western blotting. I recommend to accept the manuscript after some minor changes like

1. Line 99 Page 3; "GnT enzymes........... N-glycans (Fig A). sounds line an incomplete sentence. Please add 'GlcNAc at the' between '.....for adding' and 'branch points..." 

2. Line 104, page 3; It's 'GlcNAc' please check capitalization of the A

3. Figure 6 and 8. Please provide Blots in black and white with the ladder by their side.; Specially for figure 8 please provide more clear blots.

4. In discussion please discuss the change in ER chaperones expression and their relation with induced ER stress. 

Author Response

Reviewer 2

We much appreciate your comments and strengthening of the manuscript.

Comments and Suggestions for Authors

Hall and coworkers studies how changes in N-glycosylation influence cellular properties in NB using rat N-glycan mutant cell lines, NB_1(-Mgat1), NB_1(-Mgat2), and NB_1(-Mgat3), and the parental cell line, NB_1 in an efficient manner. They also studied the correlation between invasion property and type of N-glycan present in it and finally looked for change in ER chaperones expressions through western blotting. I recommend to accept the manuscript after some minor changes like

  1. Line 99 Page 3; "GnT enzymes........... N-glycans (Fig A). sounds line an incomplete sentence. Please add 'GlcNAc at the' between '.....for adding' and 'branch points..."

Authors: Changed to “GnT enzymes (Mgat genes) are responsible for adding branch points through the addition of GlcNAc residues to the conserved pentasaccharide of N-glycans (Fig 1A).”

  1. Line 104, page 3; It's 'GlcNAc' please check capitalization of the A

Authors: Changed GlcNac to GlcNAc

  1. Figure 6 and 8. Please provide Blots in black and white with the ladder by their side.; Specially for figure 8 please provide more clear blots.

Authors: Since we used multiplexed Western blotting, we wish to use color images to represent the fluorophores. The use of multiplexed Western blotting has been clarified in the methods and results. We have improved the blots in figure 8 as they clearly show differences between BE(2)-C and BE(2)-M17, except for Grp94.

  1. In discussion please discuss the change in ER chaperones expression and their relation with induced ER stress.

Authors: A paragraph in the discussion has been added.

Reviewer 3 Report

This study attempted to evaluate the effect of N-glycan branching of NB (neuroblastoma) on cell proliferation, migration/invasiveness and chaperone system. The mutant rat NB cell lines deficient in GnT-I, GnT-II, or GnT-III (this study) were compared with parental NB_1 cell. Compare to NB_1 cell, defects of GnT-III only show slightly difference in N-glycosylation and migration/invasiveness. However, significant differences were observed when chaperone protein levels were evaluated. The same systematic analysis was also carried out with two human NB cell lines. The defects in GnT-I and GnT-II connect with proliferation, migration and chaperone system. The manuscript is well-written and easy to follow and the results for the most part straightforward to interpret. Overall, it is a very nice and interesting study. However, there are some flaws that should be addressed.

Major comments:

(1)    The glycomic analysis did not show obvious difference between NB_1 and NB_1(-Mgat3) regarding to the bisecting GlcNAc. The effect on chaperone system is significant. Could the authors provide additional data to connect GnT-III with the changes in chaperon system (e.g., rescue)?

(2)    Is the N-glycan profile of human NB cell lines similar or different from that of rat NB cell? Are the expressions of GnT enzymes in human NB cell lines down-regulated? How to evaluate the interactions between the level high-mannose in human NB cell and cell invasiveness/migration?

Minor comments:

Line 12, explain ”NB”.

Line 104, replace “GlcNac” with “GlcNAc”. There are several “GlcNac” in the text. Please make it consistent.

Line 120, it is not the first appearance of “GlcNAc”

Line 127, Figure 1, panel B, there are four mutants. Which one was used for this study? Panel C, E-PHA lectin blotting mutant lane contained “’+”. Does it mean something? Please add protein marker to panel C. One band in Mgat3 mutant was still intense, while another was reduced compared to NB_1. Why?  Please move “Red hyphens indicate…” to (B).

Line 146, please expand “lengthening of N-glycans in NB cells promote cell proliferation rates…slowed rates”. How did the authors know the length of N-glycans of Mgat3 mutant by cell proliferation?

Line 217, please rephrase “isolated permethylated N-glycans…mutant cell lines were analyzed by ESI-MS. The permethylated N-glycans…labeling N-glycosylated protein…”.

Line 221, please rephrase “Moreover, relative abundancies…complex types of N-glycans”. Did relative abundance obtain from MS1 spectra?

Line 222, Table S1-S4. Could the S1-S4 be combined? Figures S5-S8 consisted of 4 annotated MS2. Why did the authors choose these 4 structures? Are they relevant to the mutants?

Line 227, is the result from previous or current study “electrophoretic mobility shifts of N-glycosylated Kv3.1 protein”? If it is current study, please show the relative result.

Line 232, Figure 4, the profile of NB_1 and NB_1(-Magt3) is very similar, which was also discussed by the authors (line 461). The authors believed they were the contaminants. Please clarify the contaminants (medium?). They are many agalactosylated N-glycans. Are they typical for NB-1 cell? In the following text and Figure 5, the authors described the glycan features between parental cell line (NB-1) and 3 mutants. It will be meaningful if plotting GlcNAc on C3 branch (GnT-I product)), GlcNAc on C6 branch (GnT-II product) and, especially, bisecting bisecting GlcNAc (GnT-III product). 

Line 313, Lectin “GNL” stands for? Why the GNL lectin was used for human NB cell lines not rat ones?

Line 388, “we show that the human BE(2)-C NB cell line with lengthened N-glycans was…”. Where did the result show in the manuscript?

Line 419, please add reference or experimental data to support “our examination of human NB cell lines showed that the BE(2)-C cell line which had more and less of the complex and…”.

Line 464, the two augments of “The lower level of complex N-glycans in” and “Moreover, a less active…” seem contradictory.

Author Response

Reviewer3

We wish to thank you for your comments and suggestions, as it more nicely demonstrates how the results support the conclusions.

Comments and Suggestions for Authors

This study attempted to evaluate the effect of N-glycan branching of NB (neuroblastoma) on cell proliferation, migration/invasiveness and chaperone system. The mutant rat NB cell lines deficient in GnT-I, GnT-II, or GnT-III (this study) were compared with parental NB_1 cell. Compare to NB_1 cell, defects of GnT-III only show slightly difference in N-glycosylation and migration/invasiveness. However, significant differences were observed when chaperone protein levels were evaluated. The same systematic analysis was also carried out with two human NB cell lines. The defects in GnT-I and GnT-II connect with proliferation, migration and chaperone system. The manuscript is well-written and easy to follow and the results for the most part straightforward to interpret. Overall, it is a very nice and interesting study. However, there are some flaws that should be addressed.

Major comments:

  • The glycomic analysis did not show obvious difference between NB_1 and NB_1(-Mgat3) regarding to the bisecting GlcNAc. The effect on chaperone system is significant. Could the authors provide additional data to connect GnT-III with the changes in chaperone system (e.g., rescue)?

Authors: We have replaced figure 4 spectra with those from the supplementary section which more clearly show how the bisecting N-glycans are different between the NB_1 and NB_1(_Mgat3) cell lines. Moreover, we have elaborated on the results describing figure 4 to support changes in bisecting GlcNAc N-glycans in the mutant cell line relative to the parental cell line. The figure legend and discussion has been updated as well. The NB_1(-Mgat3) cell line had a different N-glycan profile as shown is figure 4 and the abundancies of the N-glycan were different from the parental cell line. Additionally, the expression of chaperone proteins was different. Of note, N-glycans and chaperones are closely linked and therefore altering one will alter the other as can be viewed in the engineered NB cell lines and the human NB cell lines. This is part of a future study.

Is the N-glycan profile of human NB cell lines similar or different from that of rat NB cell?

Authors: Based on lectin blots of NB_1 in our current and past (Hall et al., 2018; 2020; 2021, see ref list) studies, the levels of oligomannose and complex type N-glycans are quite different between the two human NB cell lines. Also, you can view slide 29 of the supplementary data which shows reduced binding of glycoproteins by L-PHA and a greater band pattern of oligomannosylated N-glycans based on GNL binding for NB_1 versus the two human NB cell lines. The NB_1 cell in the L-PHA and GNL lectin blots are Lanes 1 and 2, respectively. We also have unpublished data and continue to collect additional data that indicates difference in N-glycan population between human and rat cell lines.  

Are the expressions of GnT enzymes in human NB cell lines down-regulated?

Authors: We have not edited genes in the two clonal cell lines from SK-N_BE(2). However, our lectin blotting data indicates differences in N-glycan populations which indicate changes in the N-glycosylation pathway.

How to evaluate the interactions between the level high-mannose in human NB cell and cell invasiveness/migration?

Authors: To clarify, the GNL blot shows the lane intensity of the glycoproteins from BE(2)-C cells is lower than those from BE(2)-M17, indicating that the earlier cells express lower levels of oligomannosylated glycoprotein. Moreover, the BE(2)-C cell line was less invasive. Our GNL blots from current and past studies are in agreement with the MS-ESI results showed that NB_1, NB_1(-Mgat1) NB_1(-Mgat2) and NB_1(-Mgat3) cell lines had different levels of oligomannose type N-glycans and that those with higher levels of this type were much more invasive. Taken together, the data supports that NB cells with higher levels of oligomannose are more invasive. 

Minor comments:

Line 12, explain ”NB”.

Authors: NB stands for Neuroblastoma. This was changed for clarity.

Line 104, replace “GlcNac” with “GlcNAc”. There are several “GlcNac” in the text. Please make it consistent.

Authors: Corrected

Line 120, it is not the first appearance of “GlcNAc”

Authors: Corrected

Line 127, Figure 1, panel B, there are four mutants. Which one was used for this study? Panel C, E-PHA lectin blotting mutant lane contained “’+”. Does it mean something? Please add protein marker to panel C. One band in Mgat3 mutant was still intense, while another was reduced compared to NB_1. Why?  Please move “Red hyphens indicate…” to (B).

Authors: There were four indel mutations identified in the isolated cell clump (cell clone) which was expanded, see lines136-137 and 588-589. Of note, each of the indels produced frame shift mutations with premature stop codons.

Removed the + from Figure 1C.

The protein marker for the lectin blot and gel have been added in Figure 1C. For a complete view, see supplementary containing Western blots and Lectin blots (Fig S1).

The overall lane intensity is the interest, not so much individual bands. The overall lane intensity of NB_1 (-Mgat3) cells is greatly reduced compared to parental NB_1 cells.

Red hyphens represent deleted nucleotides as edited in figure legend.

Line 146, please expand “lengthening of N-glycans in NB cells promote cell proliferation rates…slowed rates”. How did the authors know the length of N-glycans of Mgat3 mutant by cell proliferation?

Authors: For clarity, this has been revised to the following: Taken together, these results demonstrate that more complex type N-glycans in NB cells promote cell proliferation rates while truncated N-glycans, such as oligomannose and hybrid types, favor slowed rates (lines 163-165).

Line 217, please rephrase “isolated permethylated N-glycans…mutant cell lines were analyzed by ESI-MS. The permethylated N-glycans…labeling N-glycosylated protein…”.

Authors: N-glycans from parental and N-glycosylation mutants were permethylated and isolated, and then analyzed by ESI-MS (lines 250-252).

Line 221, please rephrase “Moreover, relative abundancies…complex types of N-glycans”. Did relative abundance obtain from MS1 spectra?

Authors: Moreover, relative abundancies of the glycan structures were determined from the MS1 and MS2 spectra for each sample performed in duplicate (Table S1-S4 and Figures S1-S8). All four cell lines contained detectable levels of oligomannose, hybrid and complex types of N-glycans.

Line 222, Table S1-S4. Could the S1-S4 be combined? Figures S5-S8 consisted of 4 annotated MS2. Why did the authors choose these 4 structures? Are they relevant to the mutants?

Authors: For clarity, the tables have not been combined.

They were selected since they were of high abundance for oligomannose, hybrid and complex types of N-glycans.

Line 227, is the result from previous or current study “electrophoretic mobility shifts of N-glycosylated Kv3.1 protein”? If it is current study, please show the relative result.

Authors: References were added.

Line 232, Figure 4, the profile of NB_1 and NB_1(-Magt3) is very similar, which was also discussed by the authors (line 461). The authors believed they were the contaminants. Please clarify the contaminants (medium?). They are many agalactosylated N-glycans. Are they typical for NB-1 cell? In the following text and Figure 5, the authors described the glycan features between parental cell line (NB-1) and 3 mutants. It will be meaningful if plotting GlcNAc on C3 branch (GnT-I product)), GlcNAc on C6 branch (GnT-II product) and, especially, bisecting bisecting GlcNAc (GnT-III product).

Authors: Figure 4 has been changed to clarify the differences between NB_1 and NB_1(-Mgat3). The text of the results and discussion have been updated as well. The revised figure nicely shows differences between the mutant cell lines and the parental cell line.

For clarity, contaminants have been removed. It was referring to the glycoproteins in FBS which may bind tight to the plasma membrane of the cells.

The levels of nongalactosylated were about 5% in the parental cell line and we agree with your observation.

The various types of N-glycans were plotted as oligomannose, hybrid and complex to show different N-glycan populations between the various NB cell lines. Further the percent of bisecting GlcNAc hybrid and complex N-glycans are stated in the text of the results, describing figure 4.

Line 313, Lectin “GNL” stands for? Why the GNL lectin was used for human NB cell lines not rat ones?

Authors: Galanthus Nivalis Lectin (GNL). GNL has binding specificity to high mannose type N-glycans, added to text of results. It has been used previously to characterize NB_1 (-Mgat1) and NB_1 (-Mgat2) cells and lectin blots showed that their glycoproteins bound much more GNL than those of NB_1, see Hall et al., PLos One. 2021 Nov8; 16(11):e0259743). It was not used for NB_1 (-Mgat3) since it was not different from NB_1.

Line 388, “we show that the human BE(2)-C NB cell line with lengthened N-glycans was…”. Where did the result show in the manuscript?

Authors: By lengthening, we are referring to more processed. For clarity, we have changed lengthened to more processed throughout the text. This was based off of the increased binding of LPHA lectin in BE(2)-C relative to BE(2)-M17. Increased binding of LPHA indicate that BE(2)-C carries more complex type N-glycan structures relative to BE(2)-M17.

Line 419, please add reference or experimental data to support “our examination of human NB cell lines showed that the BE(2)-C cell line which had more and less of the complex and…”.

Authors: This is shown in figure 7A where L-PHA binding was increased in BE(2)-C relative to BE(2)-M17.

Line 464, the two augments of “The lower level of complex N-glycans in” and “Moreover, a less active…” seem contradictory.

Authors: Having an increased diversity of complex N-glycan structures due to an inactive GnT-III could dilute the individual structures relative to more abundant structures to the point where they are not likely to be detected.

Of note, we have changed figure 4, the text of the results and discussion to support bisecting GlcNAc N-glycans were reduced in NB_1(-Mgat3) relative to NB_1.

Round 2

Reviewer 1 Report

Authors have made the requested corrections. The manuscript can be published

Author Response

Thanks!

Reviewer 3 Report

The authors addressed most comments carefully and precisely. The quality of blurry Figure 4 should be improved.

Author Response

The quality of Figure 4 was improved.